# Mixture of Balanced Information Bottlenecks for Long-Tailed Visual Recognition

**Yifan Lan**                                                  *yifanlan@hust.edu.cn*
*Huazhong University of Science and Technology*

**Xin Cai**                                                    *xincai1998@gmail.com*
*Huazhong University of Science and Technology*

**Jun Cheng**                                                  *jcheng24@hust.edu.cn*
*Huazhong University of Science and Technology*

**Shan Tan**                                                   *shantan@hust.edu.cn*
*Huazhong University of Science and Technology*

**Reviewed on OpenReview:** *https://openreview.net/forum?id=9eiALSuZGA*

## Abstract

Deep neural networks (DNNs) have achieved significant success in various applications with large-scale and balanced data. However, data in real-world visual recognition are usually long-tailed, bringing challenges to efficient training and deployment of DNNs. Information bottleneck (IB) is an elegant approach for representation learning. In this paper, we propose a *balanced information bottleneck* (BIB) approach, in which loss function re-balancing and self-distillation techniques are integrated into the original IB network. BIB is thus capable of learning a sufficient representation with essential label-related information fully preserved for long-tailed visual recognition. To further enhance the representation learning capability, we also propose a novel structure of *mixture* of multiple *balanced information bottlenecks* (MBIB), where different BIBs are responsible for combining knowledge from different network layers. MBIB facilitates an end-to-end learning strategy that trains representation and classification simultaneously from an information theory perspective. We conduct experiments on commonly used long-tailed datasets, including CIFAR100-LT, ImageNet-LT, and iNaturalist 2018. Both BIB and MBIB reach state-of-the-art performance for long-tailed visual recognition.

## 1 Introduction

With the emergence of ImageNet, COCO, and other datasets, deep neural networks (DNNs) have achieved great success in various computer vision tasks such as image classification, object detection, and image segmentation. The success of deep learning is largely due to the large-scale and balanced data in these tasks. However, real-world data are usually long-tailed (Buda et al., 2018), which means that a few classes (head classes) occupy most of the instances in the data, while most classes (tail classes) occupy only a few instances. When using standard training methods (using cross entropy as the loss function and instance balanced sampling) to train a model on a long-tailed training set, the model usually performs well for the head classes but poorly for the tail classes, which brings a great challenge to the training and deployment of DNNs (Buda et al., 2018).

Recently, many methods have been proposed to solve the problem of imbalanced data distribution. Among these methods, re-sampling and loss function re-balancing are the two most commonly used techniques. Re-sampling makes an imbalanced dataset balanced by over-sampling or under-sampling (Buda et al., 2018;

Ando & Huang, 2017). Re-balancing methods include re-weighting (Buda et al., 2018) and logits adjustment (Ren et al., 2020). Kang et al. (2019) found that although the classification performance of a model trained by standard methods is worse than that of re-balancing methods, the obtained feature space is better. They proposed to decouple the training process into two stages: learning representation and learning classifier. The decoupling training method leads to good results (Peifeng et al., 2023; Zhou et al., 2023; Zhang et al., 2021). Zhong et al. (2021) showed that partial network parameters obtained in the learning representation stage, such as parameters of Batch Normalization (BN), are not suitable for the second stage of learning classifier directly, which suggests that learning representation and learning classifier should not be regarded as two completely independent stages. Mixture-of-Experts (MoE) techniques involve the training of multiple neural networks, each specializing in handling distinct segments of a long-tailed dataset. BBN (Wang et al., 2020), RIDE (Zhou et al., 2020) and SADE (Zhang et al., 2022) serve as representations of MoE methods. However, the enhanced capabilities of MoE come at the cost of increased computational loads. Recently, Laurent et al. (2022) highlighted that, for long-tailed visual recognition, the key is not just the classification rule but the ability to learn and identify correct features.

The Information Bottleneck (IB) is an elegant approach for representation learning. It originates from the rate-distortion theory (Tishby, 1999; Tishby & Zaslavsky, 2015), and has made extraordinary progress in many tasks, such as image classification (Alemi et al., 2017), image segmentation (Luo et al., 2019), multi-view learning (Wang et al., 2021b), reinforcement learning (Goyal et al., 2019) and so on. IB rethinks what a "good" representation is: for a given task, the best representation should contain sufficient and minimal amount of information. In the IB theory, sufficiency is achieved by maximizing mutual information between the representation $z$ and the label $y$, and minimality is achieved by minimizing mutual information between the input $x$ and the representation $z$. By introducing a Lagrange multiplier $\beta$, the IB method can be optimized by minimizing

$$R_{IB} = -I(z;y) + \beta I(x;z), \quad I(a;b) = D_{KL}(P_{(a,b)} \| P_a \otimes P_b), \tag{1}$$

where $I(a;b)$ represents mutual information between $a$ and $b$, which measures the mutual dependence between $a$ and $b$. $D_{KL}$ is the Kullback–Leibler divergence, and $P_a \otimes P_b$ is the outer product distribution which assigns probability $P_a(a) \cdot P_b(b)$ to each $(a,b)$.

IB provides a new learning paradigm that naturally avoids over-fitting and enhances model robustness. To optimize the IB objective in problems involving high-dimensional variables, variants such as VIB (Alemi et al., 2017), Drop-IB (Kim et al., 2021), and Nonlinear-IB (Kolchinsky et al., 2019) are proposed. VIB is the first to indirectly optimize the variational upper bound of IB, and has become one of the most widely used IB variants for its simplicity and effectiveness.

Although IB has made a lot of progress on balanced datasets, $I(z;y)$ is still affected by label distribution when applied to long-tailed datasets. This limitation arises from the scarcity of tail class samples, making it difficult for DNNs to learn a sufficient representation $z$. Besides, mutual information is difficult to estimate in DNNs. To solve these problems, we propose a novel *balanced* information bottleneck (BIB) method in this study. We use the loss function re-balancing technique to alleviate challenges posed by label distribution. Concurrently, we implicitly optimize the information bottleneck objective by self-distillation to reserve as much information related to labels as possible in the process of information flow. To further enhance the representation learning ability, we introduce a novel framework of a mixture of multiple *balanced* information bottlenecks (MBIB). MBIB is the first network to leverage *multiple* balanced information bottlenecks, each responsible for extracting knowledge from different network layers. By optimizing multiple IB objectives simultaneously, MBIB ensures a comprehensive and effective representation learning process, leading to improved performance. We conduct experiments on benchmark datasets, including CIFAR100-LT, ImageNet-LT, and iNaturalist 2018.

The contributions of this study are summarized as follows:

1) Firstly, we propose a novel *balanced* information bottleneck (BIB) method to handle long-tailed data for real-word visual recognition.
2) Secondly, to the best of our knowledge, we propose the first network with a mixture of *multiple* balanced information bottleneck (MBIB), optimizing diverse IB objectives simultaneously to effectively learn representation and classifier in an end-to-end fashion.

3) Finally, our methods achieve state-of-the-art performance among single-expert methods and competitive performance with MoE methods while maintaining higher efficiency, as evidenced by the average classification accuracy across multiple benchmark datasets.

## 2 Related Work

### 2.1 Long-Tailed Visual Recognition

Re-sampling and loss function re-balancing are the most widely studied approaches in imbalanced classification. Re-sampling aims to construct a balanced training set, including over-sampling and under-sampling. Due to the low diversity of tail classes, over-sampling often results in models over-fitting tail classes. Under-sampling discards part of samples in head classes, which damages the diversity of head classes and decreases model generalization performance. Kang et al. (2019) proposed a progressive sampling algorithm combining over- and under-sampling to transition the distribution from an imbalanced distribution to a balanced distribution. However, the problem of re-sampling still exists.

Loss function re-balancing gives different weights to different classes or instances to re-balance the model at the level of the loss function. This strategy mainly includes re-weighting (Cui et al., 2019; Byrd & Lipton, 2019) and logits adjustment (Ren et al., 2020; Menon et al., 2021). Cui et al. (2019) proposed re-weighting the loss function with the number of effective samples. Lin et al. (2020) suggested that the model should pay more attention to hard samples by giving hard samples a large weight. On the other hand, Ren et al. (2020) and Menon et al. (2021) adjusted logits to make gradients more balanced during training. Kang et al. (2019) found that although re-balancing and re-sampling can improve the performance of imbalanced classification, they result in worse representation. Kang et al. (2019) proposed to decouple the training process into two stages. The first stage uses standard training methods to learn a good representation. In the second stage, strategies such as re-sampling, re-weighting and so on are used to fine-tune the classifier to learn a good classifier. The decoupling training approach provides a new training paradigm for long-tailed recognition. Many studies have focused on how to obtain better representation (Kang et al., 2021; Liu et al., 2021; Zhong et al., 2022) or learn a better classifier (Wang et al., 2021d;a; Zhong et al., 2021). MoE methods also have achieved a great success by effectively combining the knowledge from multiple experts. BBN (Zhou et al., 2020) introduces a two-branches network to address long-tailed recognition. RIDE (Wang et al., 2020) trains multiple experts with the softmax loss respectively and enforces a KL-divergence based loss to enhance the diversity among various experts. SADE (Zhang et al., 2022) pioneers a novel spectrum-spanned multi-expert framework and introduces an innovative expert training scheme. Distillation strategy like DiVE (He et al., 2021) and contrastive learnig like PaCo (Cui et al., 2021) have also achieved successes.

### 2.2 Information Bottleneck

IB divides the model into two parts: an encoder and a decoder. The encoder codes a random variable $x$ into a random variable $z$, and the decoder decodes the random variable $z$ into a random variable $y$. IB assumes that variables $x$, $z$, $y$ follow a Markov chain $y \leftrightarrow x \leftrightarrow z$, and $z$ is the bottleneck of the information flow. IB expects $z$ to retain information from $x$ to $y$ as much as possible while forgetting information unrelated to $y$. This means that for a given task, $z$ is the best representation in the perspective of rate-distortion theory. However, despite the beauty of the IB theory, the calculation of mutual information in IB is complicated, especially in deep learning (Slonim, 2006). To address this problem, the Variational Information Bottleneck (VIB) (Alemi et al., 2017) optimizes the IB objective by using its upper bound. According to VIB, $-I(z; y)$ and $I(x; z)$ are bounded as

$$
\begin{aligned}
-I(z; y) &\leq \mathbb{E}_{p(z,y)} - \log q(y|z) = L_{CE}(z, y), \\
I(x; z) &\leq \mathbb{E}_{p(x,z)} \log p(z|x) - \mathbb{E}_{p(z)} \log r(z) \\
&= \mathbb{E}_{p_{data}(x)}[KL(q(z|x)||r(z))].
\end{aligned}
\tag{2}
$$

Here, $\mathbb{E}_{p(z,y)}$ and $\mathbb{E}_{p(x,z)}$ is the expectation over the joint distribution $p(z, y)$ and $p(x, z)$. Similarly, $\mathbb{E}_{p_{data}(x)}$ is the expectation over the data distribution $p_{data}(x)$. $q(y|z)$, $q(z|x)$, $r(z)$ are variational approximations to $p(y|z)$, $p(z|x)$, $p(z)$, respectively. These variational distributions are introduced to make the optimization

computationally simpler by approximating the intractable true distributions. VIB is widely used for its simplicity and effectiveness. However, IB is essentially a compromise between representation and classification, and obtaining an optimal compromise point is difficult. To avoid this compromise, Tian et al. (2021) proposed a method to implicitly optimize the information bottleneck objective through self-distillation.

## 3  Method

In this section, we first formalize the problem setup and the motivation for our approach from the perspective of information theory in Section 3.1. We then introduce the proposed Balanced Information Bottleneck (BIB) framework in Section 3.2, followed by its extension, the Mixture of Balanced Information Bottlenecks (MBIB) in Section 3.3, which leverages optimization of multiple IB and improves the representation further. We also explain the effectiveness of MBIB from the perspectives from information theory and self-distillation.

### 3.1  Preliminaries

Let $D = \{(x_i, y_i)\}_{i=1}^{N}$ be a training set, where $y_i$ is the label for data $x_i$ and $K$ is the number of classes. Without loss of generality, let $n_1 > n_2 > \cdots > n_K$, where $n_i$ is the number of training samples for class $i$, hence the total number of training samples is $N = \sum_{k=1}^{K} n_k$. Unlike the long-tailed distribution of the training set, the test set follows a uniform distribution, ensuring an equal number of samples across all classes for a balanced evaluation of the model's performance in each class.

The objective is to learn a sufficient and minimal representation $z$ that preserves information relevant to the label $y$, while discarding irrelevant and redundant information in $x$. From the Information Bottleneck (IB) perspective, this corresponds to optimizing the objective $R_{IB}$ in Eq. 1.

### 3.2  Balanced Information Bottleneck (BIB)

The information bottleneck theory aims to obtain a sufficient and minimal representation of the input, in the sense of effectively characterizing the output. Let $v$ be an observation of the input $x$ extracted from an encoder such as a CNN, and $z$ be a representation encoded from $v$ by a fully connected layer, as shown in Figure 1. Considering CNNs' powerful feature extraction capability, we introduce the following assumption to enable tractable optimization of the IB objective (Tian et al., 2021):

**Assumption 3.1** (Sufficiency of Observation $v$). *The observation $v$ extracted by the encoder is assumed to retain all label-relevant information from the input $x$, i.e., $I(v; y) = I(x; y)$.*

Therefore, our problem is translated into how to find a representation $z$ that preserves the sufficient and minimal information of the observation $v$ to the label $y$.

**Theorem 3.2** (Three Sub-optimization Objectives). *Under Assumption 3.1, the Information Bottleneck (IB) objective can be decomposed into the following three sub-objectives:*

- *Maximize $I(v; y)$ — encourage $v$ to retain label-relevant information;*

- *Maximize $I(z; y)$ — encourage $z$ to be predictive of the label;*

- *Minimizing $|I(v; y) - I(z; y)|$ — encourage $z$ to contain less redundant information.*

*We need to optimize these three objectives together to get a minimal and sufficient representation $z$.*

*Proof of Theorem 3.2.* We decompose $I(v; z)$ to two terms:

$$I(v; z) = I(z; y) + I(v; z|y), \tag{3}$$

where the first term represents the information in $z$ that is related to the label, and the second term represents information that is not related to the label. To optimize the IB objective, we should maximize $I(z; y)$ while minimizing $I(v; z|y)$. Minimizing $I(v; z|y)$ is equivalent to minimizing $|I(v; y) - I(z; y)|$ (Tian

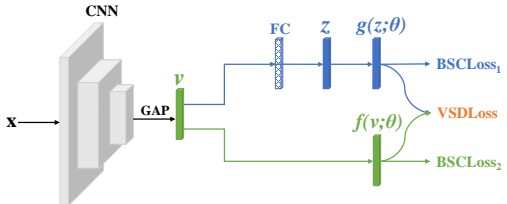

Figure 1: The network structure of BIB. FC means the Fully Connected layer and GAP means Global Average Pooling, $f(v;\theta)$ and $g(z;\theta)$ are classifiers. BSCLoss means the balanced softmax cross entropy loss ($Loss_1$ and $Loss_2$), and VSDloss means variational self-distillation loss ($Loss_3$). The output is the mean of the outputs from $f(v;\theta)$ and $g(z;\theta)$.

et al., 2021). Therefore, we can decompose the objective into three sub-optimization objectives: maximizing $I(v;y)$, maximizing $I(z;y)$ and forcing $I(z;y)$ to approximate $I(v;y)$. $\qquad\square$

To handle long-tailed data and optimize the IB objective, we propose a *balanced information bottleneck* (BIB), which integrates re-balancing techniques and self-distillation into the original network architecture. This approach mitigates data imbalance while jointly optimizing the three sub-objectives defined in Theorem 3.2 to improve the quality of representation $z$. We now formulate a loss function to address each of the three objectives from Theorem 3.2.

We start with maximizing $I(v;y)$ and $I(z;y)$. According to VIB (Alemi et al., 2017), $-I(v;y)$ and $-I(z;y)$ are bounded as $\mathbb{E}_{p(v,y)} - \log q(y|v)$ and $\mathbb{E}_{p(z,y)} - \log q(y|z)$. For a long-tailed dataset, we apply a re-balancing technique over $I(v;y)$:

$$p_s(y_i|v) \approx q_s(y_i|v) = \frac{n_i e^{f_i(v;\theta)}}{\sum_{j=1}^{K} n_j e^{f_j(v;\theta)}}, \tag{4}$$

where the subscript $s$ represents the training set distribution, $f(v;\theta)$ represents the classifier following $v$. The proof can be found in Appendix A. The loss corresponding to maximizing $I(v;y)$ becomes:

$$Loss_1 = \mathbb{E}_{p(v,y)} - \log q_s(y|v). \tag{5}$$

Similarly, to maximize $I(z;y)$, we get the loss as:

$$Loss_2 = \mathbb{E}_{p(z,y)} - \log q_s(y|z). \tag{6}$$

$Loss_1$ and $Loss_2$ are cross entropy losses. We introduce a classes weighting factor inversely proportional to the label frequency to strengthen the learning of the minority class and re-balance the losses better (Du et al., 2023). The weighting factor is:

$$w_i = \frac{K \cdot (1/d_i)^m}{\sum_{i=1}^{K} (1/d_i)^m}, \tag{7}$$

where $d_i$ is the $i$-th class frequency of the training dataset, and $m$ is a hyperparameter. Therefore, $Loss_1$ and $Loss_2$ are regarded as balanced softmax cross entropy losses (BSCLoss).

Then, to force $I(z;y)$ to approximate $I(v;y)$, we only need to ensure that $H(y|v)$ approximates $H(y|z)$. Tian et al. (2021) proved that making $q(y|v)$ approximate $q(y|z)$, i.e., minimizing the KL-divergence between $q(y|z)$ and $q(y|v)$, can effectively make $H(y|v)$ approximate $H(y|z)$. Therefore, our third loss is to minimize the $D_{KL}[q(y|v)||q(y|z)]$:

$$\begin{aligned} Loss_3 &= \mathbb{E}_{q(v|x)}[D_{KL}[q(y|v)||q(y|z)]] \\ &= \mathbb{E}_{q(v|x)}[\mathbb{E}_{q(z|v)}[q(y|v)[\log q(y|v) - \log q(y|z)]]] \\ &= \mathbb{E}_{q(v|x)}[\mathbb{E}_{q(z|v)}[-H(y|v) - q(y|v)\log q(y|z)]]. \end{aligned} \tag{8}$$

Note that this is like the method of self-distillation. To stabilize the optimization, we don't optimize $q(y|v)$; instead, we detach it from the backward propagation process. We call $Loss_3$ the variational self-distillation loss (VSDLoss). Furthermore, to mitigate the effect of the long-tailed distribution, we use class-dependent self-distillation temperatures: $q(y_i|v) = \frac{e^{f_i(v;\theta)/T_i}}{\sum_{j=1}^{K} e^{f_j(v;\theta)/T_j}}$ and $q(y_i|z) = \frac{e^{g_i(z;\theta)/T_i}}{\sum_{j=1}^{K} e^{g_j z;\theta)/T_j}}$ , where $T_i = (\frac{n_{max}}{n_i})^\gamma$, $\gamma$ is a hyperparameter, $g(v;\theta)$ represents the classifier following $v$.

The overall loss is given by

$$Loss_{BIB(v,z)} = Loss_1 + Loss_2 + \beta \cdot Loss_3, \tag{9}$$

where $\beta$ is a hyperparameter. To sum up, $Loss_1$ and $Loss_2$ are balanced cross entropy losses (BSCLoss) to maximize $I(v;y)$ and $I(z;y)$. $Loss_3$ is the variational self-distillation loss (VSDLoss) to force $I(z;y)$ to approximate $I(v;y)$. Therefore, we can optimize the IB objective implicitly by minimizing $Loss_{BIB(v,z)}$. The code of BIB loss is presented in Appendix J.

### 3.3 Mixture of Balanced Information Bottleneck (MBIB)

From the perspective of information theory, BIB assumes that the CNN-derived observation $v$ of the input $x$ satisfies $I(v;y) = I(x;y)$, as stated in Assumption 3.1, implying that $v$ preserves all mutual information between $x$ and the label $y$. However, this assumption encounters limitations due to the inherent data processing inequality (Cover & Thomas, 1991) in the information processing chain.

Specifically, consider a CNN network composed of three consecutive parts, denoted as $CNN_1$, $CNN_2$, and $CNN_3$, as illustrated in Figure 2. Let $v_1$, $v_2$, and $v_3$ denote the intermediate representations extracted from these partial networks, where $v_3$ corresponds to the full observation $v$ used in BIB (Section 3.2). We formally state the following result:

**Theorem 3.3** (Data Processing Inequality). *Let $v_1$, $v_2$, and $v_3$ be intermediate representations obtained from partial networks $CNN_1$, $CNN_2$, and $CNN_3$, respectively. Then the following inequality holds:*

$$I(v_3;y) \leq I(v_2;y) \leq I(v_1;y). \tag{10}$$

This theorem reflects the fundamental principle that, as data is progressively transformed and compressed through the network, the mutual information between the representation and the target label inevitably diminishes. As a result, Assumption 3.1 made in BIB and prior works (Tian et al., 2021) does not always hold in practice; instead, we generally have $I(v;y) \leq I(x;y)$ due to inevitable information loss along the processing chain. Consequently, the final representation $v$ ($v_3$ in Figure 2) is not a sufficient observation containing all mutual information between $x$ and $y$, and solely applying BIB between $v$ and $z$ may fail to fully exploit the label-relevant information.

One promising model enhancement strategy is leveraging the label-related information retained in $v_1$ and $v_2$, given that they contain more mutual information with $y$ than $v_3$ ,as indicated in Theorem 3.3. By applying BIB between $v_1$ and $z$ ($BIB(v_1, z)$), as well as between $v_2$ and $z$ ($BIB(v_2, z)$), we are able to conserve a greater amount of label-relevant information in $v_1$ and $v_2$. This enables feature $z$ a sufficient and comprehensive representation of $v_1$, $v_2$, and $v_3$ simultaneously and containing more label-related information, ultimately improving the model's ability to capture long-tailed label information.

Hence, we propose a novel network structure, which consists of the *mixture* of *multiple balanced information bottleneck* (i.e., MBIB), as shown in Figure 2. MBIB is a network capable of simultaneously optimizing diverse information bottleneck. The overall loss function is:

$$Loss_{MBIB} = a \cdot Loss_{BIB(v_1,z)} + b \cdot Loss_{BIB(v_2,z)} + Loss_{BIB(v_3,z)}, \tag{11}$$

where $a$ and $b$ are hyperparameters to adjust the proportions of different BIBs. Specifically, when $a = 0$ and $b = 0$, MBIB degrades to the BIB proposed in Section 3.2. As the values of $a$ and $b$ increase, the model emphasizes information from $v_1$ and $v_2$ more. Each $Loss_{BIB}$ is composed of three losses as below:

$$Loss_{BIB} = BSCLoss_1 + BSCLoss_2 + \beta \cdot VSDLoss, \tag{12}$$

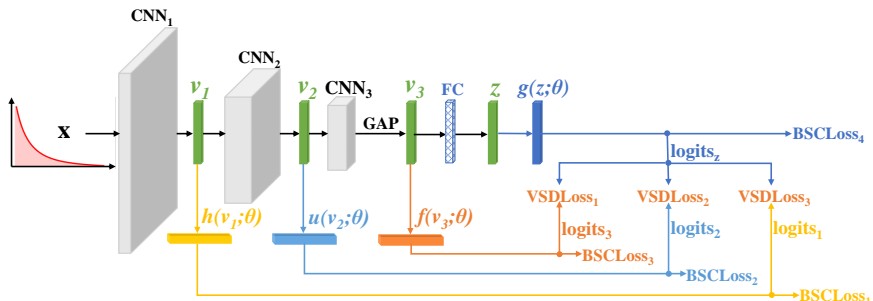

Figure 2: The network structure of MBIB. $CNN_1$, $CNN_2$ and $CNN_3$ are different parts of the CNN network. FC means the Fully Connected layer, and GAP means Global Average Pooling. $h(v_1;\theta)$, $u(v_2;\theta)$, $f(v_3;\theta)$ and $g(z;\theta)$ are classifiers. The output is the mean of the outputs from $f(v_3;\theta)$ and $g(z;\theta)$.

where there are two balanced cross entropy losses and one variational self-distillation loss.

From the perspective of self-distillation, we can further explore the effectiveness of MBIB. VSDLosses in Eqs. equation 9 and equation 12 facilitate self-distillation processes among layers at different depths and the output layer. For example, the VSDLoss in $BIB(v_1, z)$ is optimized in the following form:

$$minVSDLoss_{BIB(v_1,z)} \Leftrightarrow min\mathbb{E}_{q(v_1|x)}[D_{KL}[q(y|v_1)||q(y|z)]]. \tag{13}$$

To optimize this objective, we minimize the KL-divergence between $q(y|z)$ and $q(y|v_1)$, thereby performing distillation between $v_1$ and $z$. This process effectively integrates knowledge from $v_1$ and $z$.

Previous studies (Jin et al., 2023) have underscored that shallow parts of the deep model are able to perform better on certain tail classes, and layers of different depths excel at recognizing different classes in long-tailed data. This suggests that fusing knowledge from shallow and deep layers can better fit the long-tailed data, and VSDLosses achieve this through self-distillation.

Therefore, MBIB enables a more comprehensive output that integrates the knowledge of different shallow layers. At the same time, this approach allows shallow layers to learn from deeper ones. Consequently, this strategy can enhance the representation learning capability of the network by integrating knowledge form different depths.

## 4 Experiments

### 4.1 Datasets and Setup

**Long-Tailed CIFAR-100.** CIFAR-100 includes 60K images, of which 50K images are for training and 10K for verification. There are 100 classes in total. We use the same long-tailed version of the CIFAR-100 dataset as in Cao et al. (2019) for a fair comparison. The imbalance degree of the dataset is controlled by the Imbalanced Factor ($IF = N_{max}/N_{min}$, where $N_{max}$ represents the highest frequency and $N_{min}$ represents the lowest frequency). We conduct experiments on CIFAR-100-LT with IF of 100, 50, and 10.

**ImageNet-LT.** ImageNet-LT is a subset of long-tailed distribution sampled from ImageNet through Pareto distribution. The ImageNet-LT training set contains 115.8K images, and a total of 1000 classes. The highest frequency is 1280, and the lowest frequency is 5. There are 20 images of each class in the validation set, and 50 images in each class of the test set.

**iNaturalist 2018.** iNaturalist 2018 is a large-scale, long-tailed fine-grained dataset. iNaturalist 2018 includes 437.5K images and 8142 classes in total, with 1000 samples for the class with the highest frequency and 2 samples for the class with the lowest frequency.

According to the setting in Liu et al. (2019), we divide the dataset into three subsets according to the number of samples: Many shot (more than 100 samples), Medium shot (between 20 and 100 samples), and Few shot (less than 20 samples).

Table 1: Top-1 accuracy(%) of ResNet32 on CIFAR-100-LT with 200 epochs. Data in bold is the overall accuracy of our methods, while underlined data indicates overall performances superior to ours. This formatting is consistent in other tables.

| Method | Imbalanced Factor | | |
|---|---|---|---|
| | 100 | 50 | 10 |
| *One-Stage* | | | |
| CE | 38.3 | 43.9 | 55.7 |
| Focal Loss | 38.4 | 44.3 | 55.8 |
| LDAM-DRW | 42.0 | 46.6 | 58.7 |
| BBN | 42.6 | 47.0 | 59.1 |
| CDT | 44.3 | - | 58.9 |
| BSCE | 42.7 | 47.2 | 58.5 |
| BIB(Ours) | **44.9** | **49.8** | **60.4** |
| MBIB(Ours) | **47.5** | **51.2** | **60.9** |
| *Two-Stage* | | | |
| cRT | 41.2 | 46.8 | 57.9 |
| $\tau$-norm | 41.1 | 46.7 | 57.1 |
| KCL | 42.8 | 46.3 | 57.6 |
| TSC | 43.8 | 47.4 | 59.0 |
| SSP | 43.4 | 47.1 | 58.9 |
| *MoE* | | | |
| BBN | 42.6 | 47.0 | 59.1 |
| RIDE(2E) | 47.0 | - | - |
| RIDE(3E) | 48.0 | - | - |
| SADE | 49.8 | 53.9 | 63.6 |
| *Others* | | | |
| DiVE | 45.4 | 51.1 | 62.0 |

## 4.2 Implementation Details

**Training details on CIFAR-100-LT.** For CIFAR-100-LT, we process samples in the same way as in Cao et al. (2019). We use ResNet32 as the backbone network. To keep consistent with the previous settings (Cao et al., 2019), we use the SGD optimizer with a momentum of 0.9 and weight decay of 0.0003. We train 200 epochs for each model. The initial learning rate is 0.1, and the first five epochs use the linear warm-up. The learning rate decays by 0.01 at the $160^{th}$ and the $180^{th}$ epoch. The batch size of all experiments is 128.

**Training details on ImageNet-LT.** For ImageNet-LT, we report the results of two backbone networks: ResNet10 and ResNeXt50. We train 90 epochs for all models, using the SGD optimizer with a momentum of 0.9 and weight decay of 0.0005. For ResNet10, we use a cosine learning rate schedule decaying from 0.05 to 0 with batch size of 128. For ResNeXt50, we use a cosine learning rate schedule decaying from 0.025 to 0 with batch size of 64.

**Training details on iNaturalist 2018.** For the iNaturalist 2018, we use ResNet50 as the backbone network. The model trained 90 or 200 epochs using the SGD optimizer with a momentum of 0.9 and weight decay of 0.0001. The batch size is 64, and we use a cosine learning rate schedule decaying from 0.025 to 0.

For the setting of hyperparameters, we take $\beta$ in $\{0, 1, 2, 3, 4, 5\}$ according to different datasets. For all of the datasets, we use $a = 0.1$, $b = 0.3$ and $m = 0.1$. For CIFAR100-LT, we use $\gamma = 0$, and for ImageNet-LT and iNaturalist 2018, we use $\gamma = 0.5$. To make the results more robust, we use the mean of $f(v;\theta)$ and $g(z;\theta)$ as the final result of the test sample.

Table 2: Top-1 accuracy(%) of ResNet10 and ResNeXt50 on ImageNet-LT with 90 epochs. A ‡ or † indicates training extended to 180 or 200 epochs.

| Method | ResNet10/ResNext50 | | | |
|---|---|---|---|---|
| | Many | Medium | Few | All |
| *One-Stage* | | | | |
| CE | 57.0/65.9 | 25.7/37.5 | 3.5/7.7 | 34.8/44.4 |
| Focal Loss | 36.4/64.3 | 29.9/37.1 | 16.0/8.2 | 30.5/43.7 |
| LDAM-DRS | -/63.7 | -/47.6 | -/30.0 | 36.0/51.4 |
| LADE | -/62.3 | -/49.3 | -/31.2 | -/51.9 |
| BSCE | 53.4/62.2 | 38.5/48.8 | 17.0/29.7 | 41.3/51.4 |
| weight balancing† | -/62.0 | -/49.7 | -/41.0 | -/53.3 |
| RBL† | -/64.8 | -/49.6 | -/34.2 | -/53.3 |
| BIB(Ours) | 54.7/64.7 | 40.0/51.2 | 21.7/32.7 | **43.2/53.9** |
| MBIB(Ours) | 56.4/67.0 | 41.8/52.8 | 23.2/33.5 | **44.9/55.7** |
| *Two-Stage* | | | | |
| cRT | -/61.8 | -/46.2 | -/27.4 | 41.8/49.6 |
| $\tau$-norm | -/59.1 | -/46.9 | -/30.7 | 40.6/49.4 |
| DisAlign | -/61.5 | -/50.7 | -/33.1 | -/52.6 |
| WCDAS | 53.8/- | 41.7/- | 25.3/- | 44.1/- |
| CC-SAM | -/63.1 | -/53.4 | -/41.4 | -/55.4 |
| SRepr‡ | -/- | -/- | -/- | -/54.6 |
| *MoE* | | | | |
| BBN | -/40.0 | -/43.3 | -/40.8 | -/41.2 |
| RIDE(3E) | -/66.9 | -/52.3 | -/34.5 | 44.3/55.5 |
| SADE | -/65.3 | -/55.2 | -/42.0 | -/57.3 |
| *Others* | | | | |
| DiVE | -/64.1 | -/50.4 | -/31.5 | -/53.1 |
| PaCo | -/59.7 | -/51.7 | -/36.6 | -/52.7 |

## 4.3 Main Results

**Baseline.** We compared four mainstream approaches, including one-stage, two-stage, MoE and other approaches such as distillation and contrastive learning. The one-stage approach includes Focal loss (Lin et al., 2020), LDAM (Cao et al., 2019), BSCE (Ren et al., 2020), weight balancing(Alshammari et al., 2022), RBL(Peifeng et al., 2023), etc. The two-stage approach includes cRT (Kang et al., 2019), KCL (Kang et al., 2021), TSC (Li et al., 2021), SSP (Yang & Xu, 2020), WCDAS(Han, 2023), CC-SAM(Zhou et al., 2023), etc. The MoE approach includes BBN(Zhou et al., 2020), RIDE(Wang et al., 2020), SADE(Zhang et al., 2022). The distillation approach is DiVE(He et al., 2021). The contrastive learning method is PaCo(Cui et al., 2021). Especially, if only the results of the $v$ are concerned, BIB degenerates to BSCE.

**CIFAR-100-LT.** Table 1 compares BIB and MBIB with baseline methods on CIFAR-100-LT. As the results show, BIB achieves improvements on all imbalanced factors. MBIB is higher than most of the methods and even outperforms some MoE models. Although some MoE methods outperform MBIB, they incur much larger computational costs than our methods.

**ImageNet-LT.** Table 2 compares BIB and MBIB with baseline methods on ImageNet-LT. We conduct experiments on two backbones networks: ResNet10 and ResNeXt50. The results of RIDE, SADE and PaCo are from Zhang et al. (2023b). The experimental results show that BIB and MBIB can achieve consistent performance improvement on both small and large neural networks. The overall accuracy of MBIB is higher than all of the baseline methods except SADE.

**iNaturalist 2018.** Table 3 compares BIB with baseline methods on iNaturalist 2018. Notably, MBIB achieves the best performance for 200 training epochs among all baseline methods including MoE models.

Table 3: Top-1 accuracy(%) of ResNet50 on iNaturalist 2018.

| Method | Epoch | Many | Medium | Few | All |
|---|---|---|---|---|---|
| *One-Stage* | | | | | |
| CE | 90/200 | 72.2/75.7 | 63.0/66.9 | 57.2/61.7 | 61.7/65.8 |
| ResLT (Cui et al., 2022) | 200 | 68.5 | 69.9 | 70.4 | 70.2 |
| LADE (Hong et al., 2021) | 200 | - | - | - | 70.0 |
| BSCE (Ren et al., 2020) | 90/200 | 67.2/69.6 | 66.5/69.8 | 67.4/69.7 | 66.9/69.8 |
| weight balancing (Alshammari et al., 2022) | 200 | 71.0 | 70.3 | 69.4 | 70.0 |
| BIB(Ours) | 90/200 | 70.9/73.9 | 69.9/72.9 | 69.6/72.1 | **69.9/72.7** |
| MBIB(Ours) | 90/200 | 70.7/72.6 | 70.6/73.6 | 70.2/73.1 | **70.4/73.3** |
| *Two-Stage* | | | | | |
| cRT (Kang et al., 2019) | 90/200+10 | 69.0/73.2 | 66.0/68.8 | 63.2/66.1 | 65.2/68.2 |
| $\tau$-norm (Kang et al., 2019) | 90/200+10 | 65.6/71.1 | 65.3/68.9 | 65.9/69.3 | 65.6/69.3 |
| KCL (Kang et al., 2021) | 200+30 | - | - | - | 68.6 |
| TSC (Li et al., 2021) | 400+30 | 72.6 | 70.6 | 67.8 | 69.7 |
| DisAlign (Zhang et al., 2021) | 90/200+30 | 64.1/69.0 | 68.5/71.1 | 67.9/70.2 | 67.8/70.6 |
| WCDAS (Han, 2023) | 200+30 | 75.5 | 72.3 | 69.8 | 71.8 |
| CC-SAM (Zhou et al., 2023) | 200+30 | 65.4 | 70.9 | 72.2 | 70.9 |
| SRepr (Nam et al., 2023) | 200+20 | - | - | - | 70.8 |
| *MoE* | | | | | |
| BBN (Zhou et al., 2020) | 90/180 | 49.4/- | 70.8/- | 65.3/- | 66.3/69.6 |
| RIDE(4E) (Wang et al., 2020) | 100/200 | 70.9/70.5 | 72.4/73.7 | 73.1/73.3 | 72.6/73.2 |
| SADE (Zhang et al., 2022) | 200+5 | 74.5 | 72.5 | 73.0 | 72.9 |
| *Others* | | | | | |
| DiVE (He et al., 2021) | 90/200 | -/- | -/- | -/- | 69.1/71.7 |
| PaCo (Cui et al., 2021) | 200 | 68.5 | 72.0 | 71.8 | 71.6 |

## 4.4 Ablation Study

**How the value of $\beta$ affects our methods?** $\beta$ is a hyperparameter in the loss function, which affects the degree of information compression by the network. Figure 3 shows the impact of different $\beta$ on the overall accuracy of the CIFAR100-LT dataset. The results show that the optimal $\beta$ may be different for different imbalanced factors and methods, and we can make more fine adjustments if necessary. However, we think this may not be necessary, because simple search in $\{0, 1, 2, 3, 4, 5\}$ can already obtain satisfactory results.

**How the the value of $a$ and $b$ affects our methods?** $a$ and $b$ are hyperparameters that adjust the proportions of different BIB components within the MBIB framework. Figure 4 shows the heatmap of accuracy on the CIFAR100-LT dataset with respect to different $a$ and $b$. The results show that the values of $a$ and $b$ significantly influence the overall accuracy. Similar to the $\beta$, we can fine-tune $a$ and $b$ to get the models having better performances.

**How the the quantity of observation $v$ affects our methods?** We utilize three observation $v$ in our methods. Ablations on $a$ and $b$ show that when $a$ or $b$ is zero, 3-MBIB (three-observation MBIB) degenerates to 2-MBIB and accuracy often lowers. We also investigated 4-MBIB, 5-MBIB and 6-MBIB with results shown in Table 4. We found the quantity of observation $v$ affects MBIB's performance, improving with more $v$. However, once $v$ exceeds 3, the performance improvement diminishes and even starts to decline. We attribute this to two key factors. Firstly, 3-MBIB has fully utilized the useful information in intermediate observations, thereby limiting the additional information gained by further increasing the number of $v$. Secondly, introducing more BIB objectives complicates the optimization process, making it harder for the model to achieve effective optimization, which ultimately leads to performance decrease for 5-MBIB and 6-MBIB.

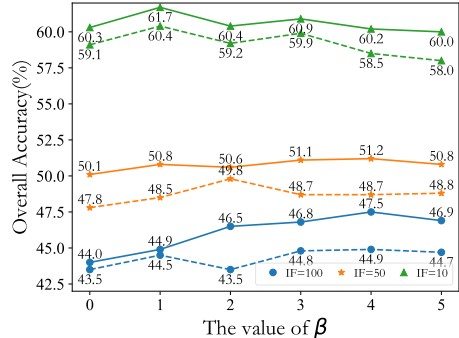

Figure 3: The impact of different $\beta$ on the overall accuracy of CIFAR-100-LT (we fixed $a = 0.1$ and $b = 0.3$). The solid lines show the results of MBIB and the dashed lines correspond to BIB.

Table 4: Accuracy of MBIB with different quantities of observation $v$. We set $a = 0$ in 2-MBIB and $a = 0.3, b = 0.3$ in the others. 4-MBIB(x) denotes the introduction of $BIB(x, z)$ to 3-MBIB.

| Quantity of $v$ | 2-MBIB | 3-MBIB | 4-MBIB(x) | 4-MBIB | 5-MBIB | 6-MBIB |
|---|---|---|---|---|---|---|
| Accuracy(%) | 45.2 | 46.6 | 46.8 | 46.9 | 46.1 | 45.4 |

Due to the page limit, the impact of different parts of BIB, as well as comparisons between BIB and BSCE are provided in Appendix B and C.

### 4.5 Analysis of the Posterior Probability Distribution

Ideally, the mean positive posterior probability of per class should be equal to 1:

$$\bar{q}(y_i|x) = \frac{1}{n_i} \sum_{j=1}^{n_i} q(y_i|x_j) = 1. \tag{14}$$

This means that the closer $\bar{q}(y_i|x)$ is to 1, the better. The experiment results reveal BIB's superiority in most classes compared to BSCE and MBIB behaves better than BIB in the tail classes. The detailed results and analysis are shown in Appendix D.

### 4.6 Analysis of the Learned Representation

A good representation should have the following characteristics: the representations of the same class are very close, and the representations between different classes are far away (Wang et al., 2021c). We can evaluate the quality of the representation by the mean of the average intra-class distance ($D_{Intra}$), the mean

Table 5: The value of $\rho$ of different methods on the CIFAR-100-LT (IF=100) testing set.

| Metric | Methods | Many | Medium | Few | All |
|---|---|---|---|---|---|
| | BSCE | 1.26 | 1.30 | 1.48 | 1.34 |
| | BSCE_MLP | 0.85 | 0.98 | 1.29 | 0.99 |
| $\rho$ | BIB_v | 0.94 | 1.06 | 1.37 | 1.06 |
| | BIB_z | 0.80 | 0.91 | 1.15 | 0.91 |
| | MBIB_v | 1.01 | 1.04 | 1.31 | 1.09 |
| | MBIB_z | **0.78** | **0.86** | **1.02** | **0.86** |

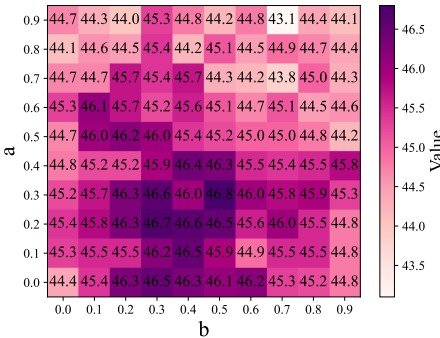

Figure 4: The impact of different $a$ and $b$ on the overall MBIB accuracy of CIFAR-100-LT (we fixed $\beta = 5$).

inter-class distance ($D_{Inter}$), and the ratio ($\rho$) between them. $D_{Intra}$, $D_{Inter}$ and $\rho$ are calculated as follows:

$$D_{Intra} = \frac{1}{K} \sum_{i=1}^{K} \frac{1}{|R_i|^2} \sum_{r_j, r_k \in R_i} \| r_j - r_k \|_2, \tag{15}$$

$$D_{Inter} = \frac{1}{K(K-1)} \sum_{i=1}^{K} \sum_{j=1, j \neq i}^{K} \| c_i - c_j \|_2, \tag{16}$$

$$\rho = \frac{D_{Intra}}{D_{Inter}}, \tag{17}$$

where $R_i$ is the representation set of class $i$, and $c_i$ is the class center of class $i$, i.e. $c_i = \frac{1}{|R_i|} \sum_{r_j \in R_i} r_j$. The better the representation, the smaller the value of $\rho$.

Table 5 compares $\rho$ obtained by different methods on the testing set. The value of $\rho$ obtained by BIB and MBIB is smaller than that of BSCE and BSCE_MLP (BSCE_MLP indicates that the network structure is the same as branch z in BIB, and the loss function is BSCE). In addition, We visualize the representation of the test set obtained by different models using t-SNE (van der Maaten & Hinton, 2008). The visualization and its analysis are shown in Appendix E.

Table 6: Efficiency comparisons (test on size $3 \times 640 \times 640$)

| Method | MBIB | SADE | RIDE(4E) |
|---|---|---|---|
| Params(k) | **486.02** | 783.86 | 1018.00 |
| FLOPs(G) | **27.93** | 40.69 | 50.98 |

### 4.7 The efficiency compared with MoE

MoE methods are often the state-of-the-art approaches in long-tailed recognition. Although they can achieve higher accuracy on some datasets than our methods, they always require larger computational resources. Therefore, MoE methods are less practical than our methods in scenarios with limited computational resources. We compare the efficiency of MBIB with MoE methods (RIDE and SADE) in Table 6.

## 5 Discussion

We have empirically observed a significant enhancement in network performance upon the introduction of BIB. Additionally, we undertook an exploration of two alternative mixture of BIB structures, as depicted in Figure 5. One structure applied sequential BIB connections between layer pairs (e.g., $BIB(v_1, v_2)$,

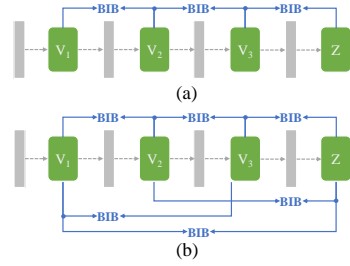

Figure 5: Two alternative multi-BIB structures.

$BIB(v_2, v_3)$, and $BIB(v_3, z)$), as shown in Figure 5(a), while the other integrated BIB connections among all feature representations as shown in Figure 5(b). We named them respectively as SE-MBIB and ALL-MBIB. However, experiments on the CIFAR100-LT dataset have revealed that both SE-MBIB and ALL-MBIB often exhibit worse performance compared to our MBIB method, as shown in Appendix F. The original intention to introduce sequential BIB connections was to propagate the optimal representation throughout the network in a cascading manner, where $v_2$ is considered a sufficient representation of $v_1$, $v_3$ is a sufficient representation of $v_2$, and $z$ is a sufficient representation of $v_3$. This would ultimately result in $z$ serving as a sufficient representation of $v_1$, consequently increasing mutual information $I(z; y)$. However, this notion is inherently less rigorous, as even if $v_2$ is a sufficient representation of $v_1$ and $v_3$ is a sufficient representation of $v_2$, $v_3$ does not necessarily constitute a sufficient representation of $v_1$. While each individual BIB operation guarantees the output as an optimal and sufficient representation of the input, the sequential BIB connections do not inherently ensure that the global output is an optimal and sufficient representation of the initial input. In other words, the representation may deviate from the initial inputs and fall into local optima instead of global optima during the sequential propagation. Thus, SE-MBIB with the sequential BIB connections appears to be less justifiable. The second structure ALL-MBIB, which introduces the improper sequential BIB connections on top of MBIB, is found to be detrimental to performance.

While our proposed methods demonstrate performance that falls slightly short of some state-of-the-art mixture of experts (MoE) models for long-tailed datasets, this presents a rich avenue for future research in BIB application. MoE methods involve the integration of multiple expert networks, each specialized in the recognition of distinct data segments. Our MBIB method could potentially be integrated into MoE frameworks. For example, MBIB may be incorporated into each expert network, enhancing the feature learning effectiveness of each expert, and thereby improving the overall performance. Inspired by two-stage methods, our network could explore staged training strategies, such as decoupling the training of the feature extractor and classifier, to optimize both components and ultimately achieve superior performance.

## 6    Conclusion

This paper proposes end-to-end learning methods named BIB and MBIB for long-tailed visual recognition based on information bottleneck theory. BIB uses self-distillation to optimize the objective and re-balance the classes, improving tail class performance without damaging head class performance. MBIB optimizes various information bottleneck within a single network simultaneously to utilize more information related to labels. Moreover, MBIB can fuse knowledge from different depths of the network to better fit the long-tailed data, further improving the model performance. Our experiments show that the quality of the feature spaces learned by BIB and MBIB is better than that of the re-balancing method like BSCE.Experiments on datasets like CIFAR100-LT, ImageNet-LT, and iNaturalist 2018 show that both BIB and MBIB perform well, even better than some recently proposed two-stage methods and MoE methods.

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

## A  Proof of the Re-Balance Technique

According to VIB, $-I(v, y)$ and $-I(z, y)$ are bounded as $\mathbb{E}_{p(v,y)} - \log q(y|v)$ and $\mathbb{E}_{p(z,y)} - \log q(y|z)$. However, when the labels are long-tailed, we need to re-balance them. Our purpose is to train an end-to-end model, that is, the output of the model is $p_t(y|v)$. According to the properties of the conditional probability, we have

$$p_s(y_i|v)p_s(v) = p_s(v|y_i)p_s(y_i), \tag{18}$$

$$p_t(y_i|v)p_t(v) = p_t(v|y_i)p_t(y_i), \tag{19}$$

where the subscript $s$ represents the training set distribution, and the subscript $t$ represents the test set distribution. Since the training set and the testing set come from the same image domain, it can be assumed that $p_s(v) = p_t(v)$ and $p_s(v|y_i) = p_t(v|y_i)$. We have

$$\frac{p_s(y_i|v)}{p_s(y_i)} = \frac{p_t(y_i|v)}{p_t(y_i)}. \tag{20}$$

Due to $p_s(y_i) = \frac{n_i}{\sum_{j=1}^{K} n_j} = \frac{n_i}{N}$ and $p_t(y_i) = \frac{1}{K}$, we have $p_s(y_i|v) \propto n_i p_t(y_i|v)$. By normalizing $p_s(y_i|v)$, we get

$$p_s(y_i|v) = \frac{n_i p_t(y_i|v)}{\sum_{j=1}^{K} n_j p_t(y_j|v)}. \tag{21}$$

We can obtain $p_t(y_i|v)$ by the output of model, so that

$$p_t(y_i|v) \approx q_t(y_i|v) = \frac{e^{f_i(v;\theta)}}{\sum_{j=1}^{K} e^{f_j(v;\theta)}}. \tag{22}$$

We can rewrite Eq. (21) as

$$p_s(y_i|v) \approx q_s(y_i|v) = \frac{n_i e^{f_i(v;\theta)}}{\sum_{j=1}^{K} n_j e^{f_j(v;\theta)}}. \tag{23}$$

Therefore, we get the first loss as

$$Loss_1 = \mathbb{E}_{p(v,y)} - \log q_s(y|v). \tag{24}$$

Similarly, for maximizing $I(z, y)$, we get the second loss as

$$Loss_2 = \mathbb{E}_{p(z,y)} - \log q_s(y|z). \tag{25}$$

## B Impact of Different Components of BIB

In order to further understand the influence of different components of the BIB loss function on the experimental results, we conducted ablation experiments on the loss function of BIB. As shown in Figure 6, when the $\beta$ is not 0, the performance of the model can be improved, which indicates that information compression by information bottleneck is conducive to long-tailed visual recognition. On the other hand, if the use of $Loss_1$ and $Loss_2$ has not been re-balanced, even if the information bottleneck is used, it will not achieve satisfactory results.

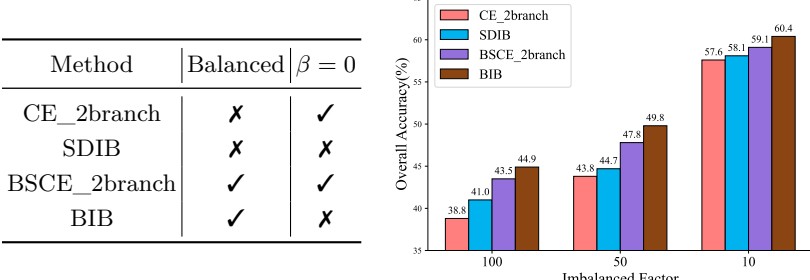

| Method | Balanced | $\beta = 0$ |
|---|---|---|
| CE__2branch | ✗ | ✓ |
| SDIB | ✗ | ✗ |
| BSCE__2branch | ✓ | ✓ |
| BIB | ✓ | ✗ |

Figure 6: Ablation results of loss function. The Balanced option is ✗ means that the method uses cross entropy loss for $Loss_1$ and $Loss_2$. The $\beta$ is 0 means that $v$ and $z$ are independent.

## C Comparison Between BIB and BSCE

From the form perspective, $Loss_1$ and $Loss_2$ are consistent with BSCE. In order to better understand our proposed BIB, we conducted extensive experiments on the ImageNet-LT with ResNet10. Table 7 shows the comparison results between BIB and BSCE. BSCE indicates that the network structure is the same as

branch $v$ in BIB, and the loss function is BSCE. BSCE_MLP indicates that the network structure is the same as branch $z$ in BIB, and the loss function is BSCE. BIB_v and BIB_z indicates the result obtained by directly using the output of $f(v;\theta)$ and $g(z;\theta)$. BIB_ensemble indicates the result obtained by the mean of $f(v;\theta)$ and $g(z;\theta)$, that is, the result we finally use. Table 7 shows that adding MLP only may hurt the performance of the model consistent with findings in Kang et al. (2019). Since IB can remove label-independent information from the representation as much as possible, the head classes performance of BIB_z has been significantly improved. At the same time, the tail classes performance has declined due to the limited number of tail class samples. However, the average performance has been greatly improved. We assume that $v$ can retain all the information in $x$, but the information will still be lost from $x$ to $v$. $v$ is upstream of $z$ in the information flow, and the improvement of the quality of $z$ will also lead to the improvement of the quality of $v$, so the performance of BIB_v has also been greatly improved. At the same time, the mean of $f(v;\theta)$ and $g(z;\theta)$ can achieve the best performance.

Table 7: Comparison between BIB and BSCE on ImageNet-LT with ResNet10.

| Method | Many | Medium | Few | All |
|---|---|---|---|---|
| BSCE | 53.4 | 38.5 | 17.0 | 41.3 |
| BSCE_MLP | 51.7 | 36.6 | 18.2 | 39.9 |
| BIB_v | 53.8 | **40.2** | **23.1** | 43.1 |
| BIB_z | 54.6 | 38.5 | 19.2 | 42.1 |
| BIB_ensemble | **54.7** | 40.0 | 21.7 | **43.2** |

## D  Analysis of the Posterior Probability Distribution

From equation 14, we can infer that the closer the mean positive posterior probability of per class is to 1, the better. Figure 7 (a) shows the mean positive posterior probability $\bar{q}(y_i|x)$ of BIB. Figure 7 (b) presents the diff (diff= $\bar{q}_1(y_i|x) - \bar{q}_2(y_i|x)$) between the posterior probability obtained by BIB and CE. The results show that CE is severely over-fitting to the head class and under-fitting to the tail class. Figure 7 (c) presents the diff between the posterior probability obtained by BIB and BSCE, revealing BIB's superiority in most classes. (d) shows the mean positive posterior probability of MBIB. (e) presents the diff between the posterior probability obtained by MBIB and BIB, which shows that MBIB behaves better than BIB in the tail classes.

## E  Analysis of the Learned Representations Showed by t-SNE

We visualize the representation of the test set obtained by different models using t-SNE, as shown in Figure 8. The visualization results show that the separability of inter-class obtained by BIB and MBIB increases, and the representations within a class are more aggregated. Both quantitative analysis and visualization results show that the quality of representations obtained by BIB and MBIB are better than others. It indicates that BIB and MBIB get better classification performance, and the representation spaces become better simultaneously, which is consistent with our expectations.

## F  Experiments on Two Other Mixture of BIB Networks

We discussed two other Mixture of BIB network in the paper: one structure applied sequential BIB connections between layer pairs (SE-MBIB), while the other integrated BIB connections among all feature representations (ALL-MBIB). In this section, we present the experiment results of the two alternative MBIB network. Figure 9 shows the performances of MBIB, SE-MBIB and ALL-MBIB on CIFAR-100-LT (IF=100). It is evident that both SE-MBIB and ALL-MBIB exhibit inferior performances compared to MBIB. The analysis can be found in the discussion section of the paper.

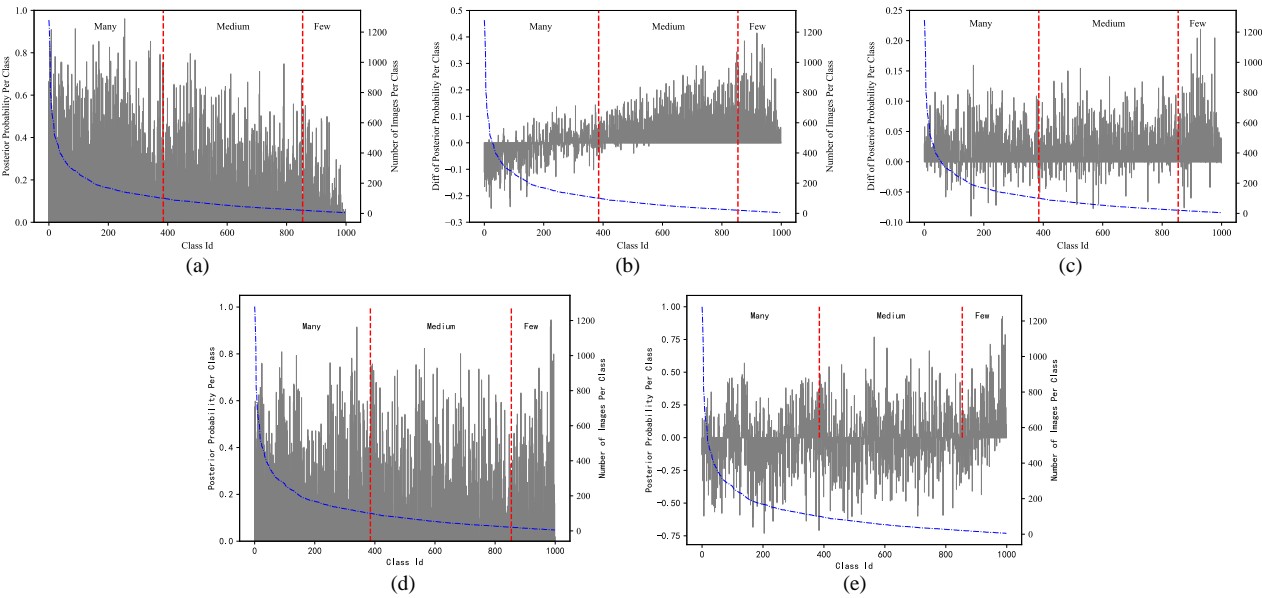

Figure 7: The mean Positive Posterior probability on ImageNet-LT with ResNet10. (a) The mean Positive Posterior probability per class of BIB. (b) Diff of the mean Positive Posterior probability per class between BIB and CE. (c) Diff of the mean Positive Posterior probability per class between BIB and BSCE.(d) The mean Positive Posterior probability per class of MBIB.(e) Diff of the mean Positive Posterior probability per class between MBIB and BIB.

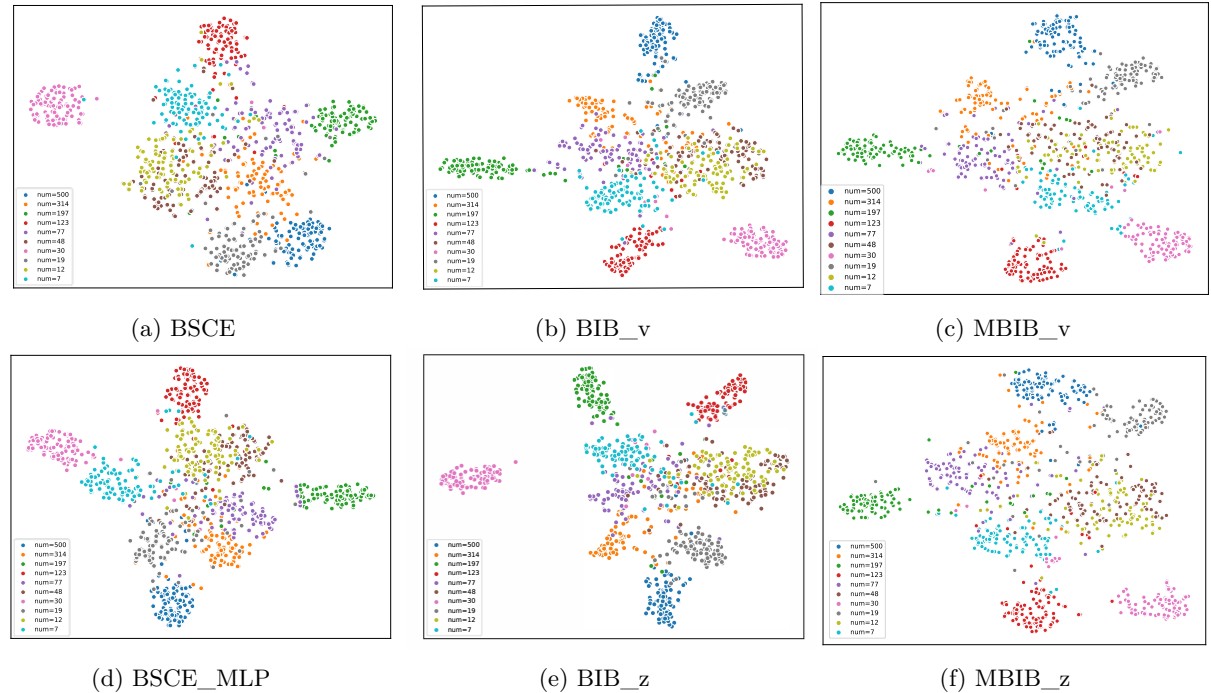

Figure 8: Visualization analysis. The t-SNE is used to visualize the test set feature space on CIFAR-100-LT (IF=100) and 10 classes are selected. The lower left corner shows the number of samples for each class during training.

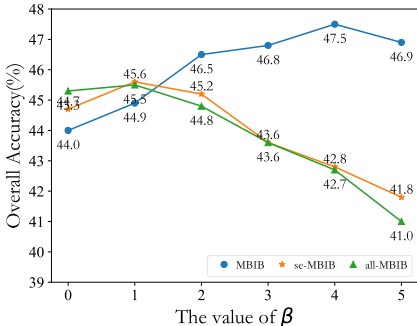

Figure 9: Overall accuracy of the three MBIB networks on CIFAR-100-LT (IF=100).

## G    Combining MBIB and MoE

As discussed in Section 5, there is significant potential for integrating MBIB into Mixture-of-Experts (MoE) frameworks. To explore this, we conducted experiments by incorporating MBIB into each expert of an MoE model, using SADE as the base architecture. The reproduced performance of SADE and the results of the combined model (SADE + MBIB) on CIFAR-100-LT (IF=100) are summarized in Table 8. We observe that this straightforward integration yields an improvement of approximately 1.6% in overall accuracy compared to the vanilla SADE model. We believe that with carefully tailored, expert-specific MBIB configurations, even greater performance gains could be achieved.

Table 8: Top-1 accuracy(%) of reproduced SADE and the comebined model (SADE + MBIB).

| Method | All | Many | Medium | Few |
|---|---|---|---|---|
| SADE | 48.7 | 62.4 | 50.2 | 30.7 |
| SADE + MBIB | 50.3 (+1.6) | 59.9 | 47.4 | 35.8 |

## H    Impact of Different Re-balancing Methods

While prior works, such as Focal Loss, have shown the effectiveness of class re-balancing losses, recent studies (Menon et al., 2021) have highlighted their limitations, including the lack of Fisher consistency. To address these issues, logit adjustment methods have been proposed (Menon et al., 2021). Furthermore, these methods can be combined, as demonstrated in Zhang et al. (2023a). As a result, we incorporate both class re-balancing methods (Eq.7) and logits adjustment methods (Eq.4) as part of the re-balancing strategy in our approach.

We also conducted further ablation studies on the re-balancing techniques of MBIB to demonstrate the importance of each re-balancing technique, with the overall accuracy shown in the Table 9. MBIB w/o class re-balancing means $m = 0$ and the weight in Eq.7 is 1 for all classes. MBIB w/o logits adjustment means removing logits adjustment in MBIB. MBIB w/o re-balancing means not using any re-balancing techniques in MBIB. The experimental results reveal that removing either re-balancing component significantly impairs performance, especially the tail (few) classes. Using only class re-balancing methods (without logit adjustment) causes a substantial 2.2% drop in overall accuracy of all classes. Similarly, employing only logit adjustment (without class re-balancing) results in a 0.9% accuracy reduction of all classes. These findings further demonstrate the necessity of incorporating both techniques jointly.

Table 9: Ablation results of re-balancing techniques.

| Method | All | Many | Medium | Few |
|---|---|---|---|---|
| MBIB w/o re-balancing | 43.5 | 62.4 | 44.6 | 20.1 |
| MBIB w/o class re-balancing | 46.1 | 65.0 | 46.6 | 23.5 |
| MBIB w/o logits adjustment | 44.8 | 63.0 | 45.8 | 22.3 |
| MBIB | 47.0 | 64.1 | 47.7 | 26.1 |

## I  Tuning of Hyperparameters

While hyperparameters $(a, b, \beta)$ do influence performance, the variation is within a reasonable range (approximately $\pm 2\%$ overall accuracy). Importantly, across most hyperparameter configurations shown in Figure 3 and Figure 4, MBIB consistently outperforms all one-stage and two-stage baselines, demonstrating the reliability and robustness of our methods.

From the perspective of information theory, the influence of hyperparameters on the performance is justifiable, and we tune the hyperparameters based on these insights:

(1) Hyperparameters $a$ and $b$ control the relative emphasis on intermediate observations $v_1$ and $v_2$. Since $v_1$ contains the most label-related information but also the most redundant information (followed by $v_2$ and then $v_3$), we observe that:

- $a$ should not be too large to avoid optimization disruption from redundant information

- $b$ can be larger than $a$ due to $v_2$ containing less redundant information than $v_1$

- The ideal configuration should satisfy $a < b < 1$

As illustrated in Figure 4, MBIB performs well when $a$ and $b$ meet this condition, confirming this theoretical expectation. We selected $a = 0.1$ and $b = 0.3$ within these satisfactory configurations.

(2) Hyperparameter $\beta$ regulates the emphasis on the IB objective optimization. As shown in Figure 3, performance generally improves when $\beta \neq 0$, and we select the value that yields the best results.

(3) For hyperparameter $m$ that regulates the class re-balancing loss, we follow established settings from prior works (Du et al., 2023).

## J  Code of BIB Loss

We present the code and comments for BIB loss in Figure 10.

```python
def BIB_loss(labels, v_logits, z_logits, sample_per_class, reduction):
    """
    Compute the Balanced Information Bottleneck (BIB) Loss.

    Args:
        labels: A int tensor of size [batch] - class labels for each sample.
        v_logits: A float tensor of size [batch, no_of_classes] - intermediate logits.
        z_logits: A float tensor of size [batch, no_of_classes] - final logits.
        sample_per_class: A int tensor of size [no_of_classes] - class frequencies.
        reduction: string. One of "none", "mean", "sum" - how to reduce the loss.

    Returns:
        loss: A float tensor - the calculated BIB Loss.
    """
    # Hyperparameter controlling the emphasis on VSDLoss
    beta = 5

    # ---------- Temperature scaling for logit adjustment ----------
    gamma = 0
    N_max, _ = torch.max(sample_per_class, dim=0)
    n = torch.reciprocal(sample_per_class / N_max)
    T = torch.unsqueeze(torch.pow(n, gamma), 0)
    temperature = T.expand_as(v_logits)

    # ---------- Variational Self-Distillation Loss (VSDLoss) ----------
    beta_logits_T = v_logits.detach() / temperature
    beta_logits_S = z_logits / temperature
    p_T = F.softmax(beta_logits_T, dim=-1)
    # KL divergence between teacher and student
    vsd_loss = (p_T * p_T.log() - p_T * F.log_softmax(beta_logits_S, dim=-1)).sum(dim=-1).mean()

    # ---------- Class re-balancing weights ----------
    label_weighting = 0.1 # Weighting factor parameter
    weights = 1.0 / (np.array(sample_per_class) ** label_weighting)
    weights = weights / np.sum(weights)
    weights = torch.FloatTensor(weights) * len(sample_per_class)

    # ---------- Balanced Softmax Cross-Entropy Loss (BSCLoss) ----------
    spc = sample_per_class.type_as(v_logits)
    spc = spc.unsqueeze(0).expand(v_logits.shape[0], -1)

    # Apply logit adjustment
    v_logits = v_logits + spc.log() # Logit adjustment based on class frequencies
    z_logits = z_logits + spc.log() # Logit adjustment based on class frequencies

    # Calculate BSCLoss
    loss1 = F.cross_entropy(input=v_logits, target=labels, reduction=reduction, weight=weights)
    loss2 = F.cross_entropy(input=z_logits, target=labels, reduction=reduction, weight=weights)

    # ---------- Combined loss ----------
    loss = loss1 + loss2 + beta * vsd_loss
    return loss
```

Figure 10: Python code of BIB Loss.

