# OpenReview forum: "Mixture of Balanced Information Bottlenecks for Long-Tailed Visual Recognition"
_TMLR — Accepted by TMLR_

### Review · Reviewer_UDAS · 2025-03-04

**Summary Of Contributions:**

This paper proposed a Neural Network model for Long-Tailed recognition task, where the information bottleneck method is introduced in the Long-Tailed problem, two losses BIB/MBIB are introduced. Performance of these models on CIFAR100-100-LT, ImageNet-LT and iNaturalist datasets are reported to justify the their effectiveness.

**Audience:**

Yes

**Claims And Evidence:**

No

**Requested Changes:**

From the perspective of machine learning research, I am not very convinced by the theoretical validity and the empirical effectiveness of the model. My first recommendation to the authors is considering resubmit this manuscript to a computer vision conference/journal. If that is not preferable, please consider:

1. Define the notations before using them. Formally describe the assumptions, theorems and proofs used and justify the theoretical contribution of the model.

2. Provide theoretical evidence to justify BIB/MBIB model (e.g. a theoretical model that introduce $m$, $\gamma$ naturally instead of only intuitively like “to mitigate the effect of the long-tailed distribution”).

3. Provide a convincing and effective hyperparameter tuning technique that search for $a,b,\beta,v,m$ so that the performance looks reliable.

**Strengths And Weaknesses:**

Pros:
1. The text and figures are easy to understand

2. Many experiment settings are tested

Cons:
1. The writing style of the paper seems more suitable to fit in the computer vision venue rather than the machine learning venue — the notations are usually used without definition (e.g. what is $E_{p(z,y)}$ in Eq. 2?);   The main contribution is the carefully designed new loss to train Neural Network models that consists of many terms (e.g. Eq. 11, 12) and the modification of the Neural Network (Fig. 2).

2. The performance of the model is not very convincing. The proposed model introduces too many hyperparameters ($a, b, \beta, v, m$), and the corresponding accuracy can differ with up to 2% (Fig. 4). This character diminish the reliablility of the model — the reported superior performance (Tab.1, 2, 3) looks more likely to be achieved by careful hyperparameter tuning.

3. The theoretical contribution of the model is not very strong — the introduced “re-balance technique” is just the re-weighting method that have been used widely in previous Long-Tailed models; Eq. 7 is introduced directly without further analysis, with an extra hyperparameter $m$;  The proposed MBIB loss is a simple augmentation of BIB loss over intermediate layer output of the NN model, without further theoretical analysis.

---

> ### Author Response · Authors · 2025-04-30
> **Response to Reviewer UDAS 1/2**
>
> Thank you for your constructive and valuable suggestions.
>
> **w1. Writing Style and Notation Definition**
>
> We thank the reviewer for this valuable feedback. We acknowledge that some parts of the current manuscript may reflect a writing style more common in computer vision venues. To address this, we have revised the paper to adopt a more rigorous ML-style presentation: we have ensured all notations are clearly introduced, and we have restructured Sections 3 to more formally describe the assumptions, objectives, and derivations.
>
> While the core contribution involves a novel loss formulation, we emphasize that its design is not merely heuristic or empirical. Instead, the loss is theoretically derived from the Information Bottleneck (IB) theory and KL-based mutual information approximations, as further elaborated in our response to w3. We have revised Section 3 to highlight these theoretical connections more clearly.
>
> We believe our work sits at the intersection of computer vision and machine learning. While the practical design aligns with vision tasks, the underlying contributions are grounded in machine learning theory, particularly in information theory and self-distillation. Furthermore, our approach addresses key challenges emphasized in the TMLR call for papers, including “experimental and/or theoretical studies yielding new insight into the design and behavior of learning in intelligent systems” and “new algorithms with sound empirical validation, optionally with justification of theoretical, psychological, or biological nature.” We appreciate the reviewer’s comment and have revised the manuscript accordingly to better reflect the theoretical foundation of our work.
>
> **w2. Hyperparameter Sensitivity**
>
> While hyperparameters ($a, b, \beta$) do influence performance, the variation is within a reasonable range (approximately $\pm2\%$ overall accuracy). Importantly, across most hyperparameter configurations shown in Figure 3 and Figure 4, MBIB consistently outperforms all of one-stage and two-stage baselines, demonstrating the reliability and robustness of our methods.
>
> From the perspective of information theory, the influence of hyperparameters on the performance is justifiable, and we tune the hyperparameters based on these insights:
>
> 1) Hyperparameters $a$ and $b$ control the relative emphasis on intermediate observations $v_1$ and $v_2$. Since $v_1$ contains the most label-related information but also the most redundant information (followed by $v_2$ and then $v_3$), we observe that:
>    - $a$ should not be too large to avoid optimization disruption from redundant information
>    - $b$ can be larger than $a$ due to $v_2$ containing less redundant information than $v_1$
>    - The ideal configuration should satisfy $a < b < 1$
>
> As illustrated in Figure 4, MBIB performs well when $a$ and $b$ meet this condition, confirming this theoretical expectation. We selected $a = 0.1$ and $b = 0.3$ within these satisfactory configurations.
>
> 2) Hyperparameter $\beta$ regulates the emphasis on the IB objective optimization. As shown in Figure 3, performance generally improves when $\beta \neq 0$, and we select the value that yields the best results.
>
> 3) For hyperparameter $m$ that regulates the class re-balancing loss, we follow established settings from prior works [A].
>
> We have presented our analysis of hyperparameters tuning in Appendix I.

---

> ### Author Response · Authors · 2025-04-30
> **Response to Reviewer UDAS 2/2**
>
> **w3. Theoretical Contribution of Our Approach**
>
> Although re-balancing techniques themselves are established, we systematically embed re-balancing methods into the Information Bottleneck framework to improve representation under imbalanced distributions. Furthermore, MBIB is not merely an augmentation of BIB over intermediate layer output of the model; it is theoretically motivated by both information theory and self-distillation.
>
> From the perspective of information theory. It reveals and addresses a key limitation in previous works (e.g., BIB, VSD [B]) that aimed to optimize the Information Bottleneck (IB) objective but implicitly assumed that the intermediate observation $v$ derived from CNNs was sufficient to contain all label-related information. However, according to the data processing inequality [C], label-related information inevitably diminishes during the information flow through the network.
> To mitigate this issue, we proposed MBIB, which optimizes multiple Information Bottleneck objectives across different intermediate observations ($v_1$, $v_2$, $v_3$) of the network because they contain more label-related information during the information compression process according to the information theory [C]. Thus, MBIB refines the information flow and enhance the representation through a multi-level IB optimization strategy from the viewpoint of information theory.
>
> Moreover, our findings are consistent with the prior work [D], which suggests that integrating knowledge from layers at different depths benefits long-tailed recognition tasks, as layers of different depths excel at recognizing different classes in long-tailed data. MBIB merges knowledge across different layers through self-distillation, aligning with this perspective.
>
> The design of MBIB, though simple, is theoretically motivated and leads to consistent empirical improvements. We have highlighted these theoretical motivations and analysis in Section 3.3.
>
> [A] Du, F., Yang, P., Jia, Q., Nan, F., Chen, X., & Yang, Y. (2023). Global and local mixture consistency cumulative learning for long-tailed visual recognitions. In Proceedings of the IEEE/CVF conference on computer vision and pattern recognition (pp. 15814-15823).
>
> [B] Tian, X., Zhang, Z., Lin, S., Qu, Y., Xie, Y., & Ma, L. (2021). Farewell to mutual information: Variational distillation for cross-modal person re-identification. In Proceedings of the IEEE/CVF Conference on Computer Vision and Pattern Recognition (pp. 1522-1531).
>
> [C] Cover, T. M. (1999). Elements of information theory. John Wiley & Sons.
>
> [D] Jin, Y., Li, M., Lu, Y., Cheung, Y. M., & Wang, H. (2023). Long-tailed visual recognition via self-heterogeneous integration with knowledge excavation. In Proceedings of the IEEE/CVF conference on computer vision and pattern recognition (pp. 23695-23704).

---

### Review · Reviewer_xwiL · 2025-03-24

**Summary Of Contributions:**

This paper introduces the Balanced Information Bottleneck (BIB) approach, which addresses challenges posed by long-tailed visual recognition data by integrating loss function re-balancing and self-distillation techniques. Additionally, it proposes the Mixture of Multiple Balanced Information Bottlenecks (MBIB), achieving state-of-the-art performance on long-tailed datasets.

**Audience:**

Yes

**Claims And Evidence:**

Yes

**Requested Changes:**

1. To enhance the credibility of the experimental results, it is essential to include standard deviation values in Tables 1 and 2. This addition will provide insights into the stability and reliability of the proposed methods.


2. Enhance the clarity of the paper by providing a more detailed explanation of the algorithm implementation. Consider including a pseudo code snippet to elucidate the workings of the proposed approach and reduce ambiguity stemming from approximations.


3. Address the lack of theoretical explanations regarding the benefits of introducing the additional loss term. Clearly articulate the rationale behind incorporating this term and elucidate how it contributes to the overall effectiveness of the method in handling long-tailed visual recognition tasks.

**Strengths And Weaknesses:**

Strengths

1. By integrating loss function re-balancing and self-distillation techniques into IB, the proposed approach offers a unique and effective solution for representation learning in the presence of imbalanced data distributions.

2. Through experiments on standard long-tailed datasets like CIFAR100-LT, ImageNet-LT, and iNaturalist 2018, the proposed BIB and MBIB models achieve state-of-the-art performance, showcasing some effectiveness of the approach in handling long-tailed visual recognition tasks.


Weaknesses

1. The experimental results in Tables 1 and 2 donnot contain standard deviation making the stability of the proposed methods questionable.

2. Writing should be much clearer. For example, what is the actual implementation of the algorithm, as many approximations are made. Please provide a pseudo code for instance.

3. Lack of theoretical explanations for the benefit of introducing the additional loss term.

---

> ### Author Response · Authors · 2025-04-30
> **Response to Reviewer xwiL**
>
> Thank you for your constructive and valuable suggestions.
>
> **w1. Missing Standard Deviations in Tables**
>
> Thank you for pointing this out. We have now reported the standard deviation values of overall accurac (%) over five independent runs with different random seeds on both CIFAR-100-LT and ImageNet-LT, as shown below. Preliminary results indicate that the performance fluctuations on CIFAR-100-LT are within $\pm$ 0.5%, and within $\pm$ 0.3% on ImageNet-LT, demonstrating the stability and robustness of our method.
>
> *Table 1. Overall accuracy (%) with standard deviations on CIFAR-100-LT for different imbalance factors.*
>
> | Imbalanced Factor            | 100                   | 50 | 10 |
> |-------------|:---------------------:|:----:|:------:|
> | BIB    | 44.6 $\pm$ 0.4    | 49.5 $\pm$ 0.2   | 60.5 $\pm$ 0.4   |
> | MBIB   | 46.8 $\pm$ 0.5    | 51.4 $\pm$ 0.4   | 61.1 $\pm$ 0.3   |
>
> *Table 2. Accuracy (%) with standard deviations on ImageNet-LT across different class groups.*
>
> |             | All                   | Many | Medium | Few |
> |-------------|:---------------------:|:----:|:------:| :------:|
> | BIB (Ours)   | 43.3 $\pm$ 0.2/ 53.8 $\pm$ 0.2       | 54.7 $\pm$ 0.1 / 64.6 $\pm$ 0.2        | 40.2 $\pm$ 0.3 / 51.1 $\pm$ 0.2          | 21.8 $\pm$ 0.3 / 32.5 $\pm$ 0.2      |
> | MBIB (Ours)  | 44.8 $\pm$ 0.3 / 55.8 $\pm$ 0.2       | 56.3 $\pm$ 0.2 / 67.0 $\pm$ 0.1        | 41.7 $\pm$ 0.3 / 52.8 $\pm$ 0.2         | 23.3 $\pm$ 0.3  / 33.5 $\pm$ 0.2       |
>
>
> **w2. Clarification of Implementation Details**
>
> We agree that adding code of our loss would help readers better understand the approach. We have included a concise python code snippet with detailed comments in the Appendix J describing the calculation of BIB loss.
>
> **w3. Theoretical Explanation of Additional Loss**
>
> The benefits of introducing the VSDLoss can be explained from the perspectives of infomration theory and self-distillation.
>
> From the perspective of information theory, the additional loss (VSDLoss) minimizes the KL-divergence between $q(y|v)$ and $q(y|z)$, which, according to Tian et al. [A], approximates minimizing the gap between $I(z; y)$ and $I(v; y)$. This implicitly optimizes the IB objective. Consequently, VSDLoss enhances label-related information retention in the feature $z$ and improves the quality of $z$, which has been shown to be crucial for long-tailed visual recognition in recent works [B].
>
> Moreover, the effectiveness of VSDLoss term can also be interpreted from the viewpoint of self-distillation. Recent works [C] proved that integrating knowledge from layers at different depths benefits long-tailed recognition tasks, as layers of different depths excel at recognizing different classes in long-tailed data. We observe that
>
> $\min \text{VSDLoss}_{(v, z)} \Leftrightarrow \min KL(q(y|v) \| q(y|z))$,
>
> which corresponds to performing self-distillation between the intermediate observation $v$ and the final feature $z$. This process effectively merges the knowlwdge of these two layers at different depths, thus improving the performance on long-tailed data.
>
> We have highlighted these theoretical explanations in Section 3.
>
> [A] Tian, X., Zhang, Z., Lin, S., Qu, Y., Xie, Y., & Ma, L. (2021). Farewell to mutual information: Variational distillation for cross-modal person re-identification. In Proceedings of the IEEE/CVF Conference on Computer Vision and Pattern Recognition (pp. 1522-1531).
>
> [B] Laurent, T., von Brecht, J. H., & Bresson, X. (2022). Long-tailed learning requires feature learning. arXiv preprint arXiv:2205.14553.
>
> [C] Jin, Y., Li, M., Lu, Y., Cheung, Y. M., & Wang, H. (2023). Long-tailed visual recognition via self-heterogeneous integration with knowledge excavation. In Proceedings of the IEEE/CVF conference on computer vision and pattern recognition (pp. 23695-23704).

---

### Review · Reviewer_BXxd · 2025-04-16

**Summary Of Contributions:**

The manuscript introduces a novel approach—Balanced Information Bottleneck (BIB) and mixture of BIB—to tackle long-tailed visual recognition problems, where data follow highly imbalanced distributions that challenge modern deep networks.

The authors leverage concepts from the Information Bottleneck (IB) framework, augmenting it with two key ideas arriving at BIB: Loss Function Re-Balancing and Self-Distillation for Preserving Label InformationBuilding on BIB, the paper further proposes Mixture of Balanced Information Bottlenecks (MBIB), which applies IB-based constraints at multiple levels of the network’s feature hierarchy.

Experimental results on well-known benchmarks (CIFAR-100-LT, ImageNet-LT, and iNaturalist 2018) show that the proposed methods (BIB and MBIB):
- Achieve competitive or superior accuracy compared to single-expert baselines,
- Competitive with some mixture-of-experts (MoE) solutions while using fewer parameters and FLOPs,
- Demonstrate improved learned representations, as evidenced by feature-space metrics and visualizations.

**Audience:**

Yes

**Claims And Evidence:**

No

**Requested Changes:**

1.	Critical
- Ablation of loss: Provide a more detailed ablation on if both re-balancing techniques are needed.
- MoE performance comparison: Building on top of MoE and showing improvement could be more convincing.
- Consider Non-ResNet Architectures: Brief results or references to confirm that the gains hold across transformer-based or MobileNet-like backbones would make the claims more comprehensive.
2.	Minor / Enhancements
- Further Discussion on novelty of MBIB

**Strengths And Weaknesses:**

1.	Strengths
- Well written: The paper is well written with sufficient background information to allow readers appreciate the motivation of the work.
- Theoretical Grounding: The approach is well-rooted in Information Bottleneck theory, using it to systematically address imbalances in label distributions.
- Ample Empirical Evidence: Experiments demonstrate improved classification accuracy across multiple benchmarks and confirm representation improvements via distance metrics (e.g., inter-class vs. intra-class distances) and t-SNE plots.
2.	Weaknesses
- Importance of re-balance: The BIB implementation introduced two weighting scheme, one in Eqn (4) scaling the softmax and one in Eqn (7) scaling loss. Many prior work, such as Focal Loss [A], have found re-balancing class losses sufficient. Is Eqn (4) necessary? How effective is this re-balancing?
- MoE performance: MoE outperforms the proposed approach. Why not combining these techniques?
- Limited Exploration of Other Architectures/Setup: While the main experiments are thorough, the paper predominantly uses ResNet-based architectures. Demonstrating generalizability to other architectures (e.g., Vision Transformers) might improve the paper’s scope. Similar argument goes for SGD optimizer.
- Novelty of MBIB: Using per-layer supervision has been a common technique is many representations literature, such as segmentation [B]. What is the novelty here?
3.	Minor
- Section 3.1 the notation `i` is abused, representing both sample i (xi, yi) and class I (ni).

[A] Lin, T. Y., Goyal, P., Girshick, R., He, K., & Dollár, P. (2017). Focal loss for dense object detection. In Proceedings of the IEEE international conference on computer vision (pp. 2980-2988).

[B] Ronneberger, O., Fischer, P., & Brox, T. (2015). U-net: Convolutional networks for biomedical image segmentation. In Medical image computing and computer-assisted intervention–MICCAI 2015: 18th international conference, Munich, Germany, October 5-9, 2015, proceedings, part III 18 (pp. 234-241). Springer international publishing.

---

> ### Author Response · Authors · 2025-04-30
> **Response to Reviewer BXxd 1/2**
>
> Thank you for your constructive and valuable suggestions.
>
> **w1. Importance of re-balancing Techniques**
>
> We appreciate the insightful question. While prior works, such as Focal Loss, have shown the effectiveness of class re-balancing losses, recent studies [A] have highlighted their limitations, including the lack of Fisher consistency. To address these issues, logit adjustment methods have been proposed. Furthermore, these methods can be combined, as demonstrated in [B]. As a result, we incorporate both class re-balancing methods (Eq. (7)) and logits adjustment methods (Eq. (4)) as part of the re-balancing strategy in our approach.
>
> We also conducted further ablation studies on the re-balancing techniques of MBIB， with the overall accuracy of all classes shown in the table below. MBIB w/o class re-balancing means $m = 0$ and the weight in Eq. (7) is 1 for all classes. MBIB w/o logits adjustment means removing logits adjustment in MBIB. MBIB w/o re-balancing means not using any re-balancing techniques in MBIB. The experimental results reveal that removing either re-balancing component significantly impairs performance. Using only class re-balancing methods (without logit adjustment) causes a substantial 2.2% drop in overall accuracy. Similarly, employing only logit adjustment (without class re-balancing) results in a 0.9% accuracy reduction. We have included a detailed discussion of these results in Appendix H, where the accuracy of each class is presented.
>
> | Method                     | MBIB w/o re-balancing | MBIB w/o class re-balancing | MBIB w/o logits adjustment | MBIB |
> |----------------------------|:---------------------:|:---------------------------:|:--------------------------:|:----:|
> | Overall Accuracy (%)                        | 43.5                  | 46.1                        | 44.8                       | 47.0 |
>
> **w2. MoE Performance and Combination**
>
> Thank you for the insightful suggestion. As noted in Section 5, we have discussed the potential of integrating MBIB into Mixture-of-Experts (MoE) frameworks as a promising direction for future research. To explore this, we conducted preliminary experiments by directly incorporating MBIB into each expert of a MoE model, using SADE as the base architecture. The reproduced performance of SADE and the results of the combined model (SADE + MBIB) on CIFAR-100-LT (IF=100) are summarized below. This straightforward integration yields an improvement of approximately 1.6% in overall accuracy compared to the vanilla SADE model. We believe that with carefully tailored, expert-specific MBIB configurations, even greater performance gains could be achieved. We have added a discussion of this integration potential in the Appendix G of revision.
>
> | Method      | All                   | Many | Medium | Few |
> |-------------|:---------------------:|:----:|:------:|:---:|
> | SADE        | 48.7                  | 62.4 | 50.2   | 30.7|
> | SADE + MBIB | 50.3 (+1.6)*          | 59.9 | 47.4   | 35.8|
>
> **w3. Limited Exploration of Other Architectures and Optimizers**
>
> Following the practice of most prior works on long-tailed recognition [A][B][E], we mainly adopted ResNet and ResNeXT architectures and SGD optimizer to ensure fair comparison. Importantly, MBIB is based on information theory and mainly modifies the loss function without relying on any specific architectural designs, making it theoretically applicable to a variety of backbones.
>
> We acknowledge that we did not include experiments on Vision Transformer (ViT). This is primarily because our implementation is based on the open-source codebase provided by previous long-tailed learning works [F], which does not include support for ViT models. Due to time constraints, we were unable to integrate ViT into our current framework, and we sincerely apologize for this limitation. We will consider incorporating such experiments in future work to further demonstrate the generalizability of our method.

---

> ### Author Response · Authors · 2025-04-30
> **Response to Reviewer BXxd 2/2**
>
> **w4. Novelty of MBIB**
>
> We thank the reviewer for the opportunity to clarify. Although per-layer supervision has been explored in segmentation literature, MBIB fundamentally differs in both motivation and formulation. Specifically, MBIB reveals and addresses a key limitation in previous works (e.g., BIB, VSD [C]) that aimed to optimize the Information Bottleneck (IB) objective but implicitly assumed that the intermediate observation $v$ derived from CNNs was sufficient to contain all label-related information. However, according to the data processing inequality [D], label-related information inevitably diminishes during the information flow through the network.
>
> To mitigate this issue, we propose MBIB, which optimizes multiple Information Bottleneck objectives across different intermediate observations ($v_1$, $v_2$, $v_3$) of the network, as they contain more label-related information during the information compression process according to the information theory [D]. Thus, MBIB refines the information flow and enhance the representation through a multi-level IB optimization strategy from the perspective of information theory, rather than merely adding per-layer supervision.
>
> Moreover, our findings are consistent with the prior work [E], which suggests that integrating knowledge from layers at different depths benefits long-tailed recognition tasks because layers of different depths excel at recognizing different classes in long-tailed data. MBIB merges knowledge across different layers through self-distillation, naturally complementing this perspective from the viewpoint of information theory.
>
> **w5. Minor - Notation Conflict in Section 3.1**
>
> We apologize for the confusion. We have revised this notation to avoid overloading "$i$" for both samples and classes.
>
> [A] Menon, A. K., Jayasumana, S., Rawat, A. S., Jain, H., Veit, A., & Kumar, S. (2020). Long-tail learning via logit adjustment. arXiv preprint arXiv:2007.07314.
>
> [B] Zhang, Y., Kang, B., Hooi, B., Yan, S., & Feng, J. (2023). Deep long-tailed learning: A survey. IEEE transactions on pattern analysis and machine intelligence, 45(9), 10795-10816.
>
> [C] Tian, X., Zhang, Z., Lin, S., Qu, Y., Xie, Y., & Ma, L. (2021). Farewell to mutual information: Variational distillation for cross-modal person re-identification. In Proceedings of the IEEE/CVF Conference on Computer Vision and Pattern Recognition (pp. 1522-1531).
>
> [D] Cover, T. M. (1999). Elements of information theory. John Wiley & Sons.
>
> [E] Jin, Y., Li, M., Lu, Y., Cheung, Y. M., & Wang, H. (2023). Long-tailed visual recognition via self-heterogeneous integration with knowledge excavation. In Proceedings of the IEEE/CVF conference on computer vision and pattern recognition (pp. 23695-23704).
>
> [F] Ren, J., Yu, C., Ma, X., Zhao, H., & Yi, S. (2020). Balanced meta-softmax for long-tailed visual recognition. Advances in neural information processing systems, 33, 4175-4186.

---

> ### Comment · Reviewer_BXxd · 2025-05-01
> **Response to authors**
>
> `w1`. It sounds like the proposed the loss was studied before and thus not a novelty of this paper. Is this the case?
>
> `w2`. The reported result in table for SADE is 72.9, which is different than the above. Am I missing something?

---

> > ### Author Response · Authors · 2025-05-03
> > **Response to Reviewer BXxd**
> >
> > **w1. Novelty of Losses**
> >
> > Thank you for this important point. While the re-balancing techniques and VSD method (to optimize the IB objective) has appeared separately in prior works, our contribution lies in systematically embedding these mechanisms to enhance representation on long-tailed data, which to our knowledge has not been explored before. Our method transcends a simple combination of existing techniques. Rather, we conducted a thorough analysis of the fundamental challenge in long-tailed learning—representation quality—and deliberately applied the Information Bottleneck framework to improve feature representations while simultaneously addressing data imbalance through carefully designed re-balancing strategies. Moreover, from the perspectives of information theory and self-distillation, we propose MBIB to address the limitations of prior IB optimization methods such as VSD,further improving the representation and preserving more label-relevant information.
> >
> > **w2. Clarification on SADE result**
> >
> > Thank you for raising this point. The 72.9 result you mentioned refers to SADE’s performance on the **iNaturalist 2018 dataset** (as reported in Table 3 of the main paper), while the 48.7 reported in our rebuttal reflects our own reproduction of SADE on the **CIFAR-100-LT (IF=100) dataset**, which corresponds to Table 1 in the main paper.

---

### Author Response · Authors · 2025-04-30
**Summary of Paper Revision**

We sincerely thank all reviewers for their valuable and constructive feedback. We have carefully revised the manuscript and made several improvements based on the comments and suggestions. Below, we summarize the major changes in the revised version:

- We clarified the notations in the paper, including explicit definitions of variables, and improved the overall mathematical rigor, as suggested by *Reviewer UDAS*.

- We added Assumption 3.1, Theorem 3.2, and the Proof in Section 3.2, as well as Theorem 3.3 in the Section 3.3, to formally justify the theoretical contribution of the model, as suggested by *Reviewers xwiL* and *UDAS*.

- We reorganized the writing style of Section 3 to better align with machine learning standards by incorporating formal definitions, clearer derivations, and theorem-proof structures, addressing *Reviewer UDAS*'s concern about writing style.

- We provided an ablation study in Appendix H comparing the effects of different re-balancing terms (Eq. (4) and Eq. (7)) to validate the necessity of both components, as requested by *Reviewer BXxd*.

- We added a discussion and preliminary results in Appendix G showing how MBIB can be integrated with a Mixture-of-Experts (MoE) model (e.g., SADE), as suggested by *Reviewer BXxd*.

- We added a detailed discussion and analysis of hyperparameter selection and tuning in Appendix I, including theoretical justifications and empirical observations, as requested by *Reviewer UDAS*.

- We added Python code to Appendix J to clarify the implementation details of BIB loss, and improved clarity in descriptions of algorithmic approximations, addressing *Reviewer xwiL*'s request.

We hope that these changes address the reviewers' concerns and further improve the clarity, rigor, and impact of our work. Thank you again for your thoughtful reviews.

---

### Decision · Action_Editor_ecp5 · 2025-06-19

**Recommendation:** Accept with minor revision

**Additional Comments:**

The authors are encouraged to further revise and improve the paper, considering reviewers' suggestions, especially improving the justification and presentation of the paper to resolve the reviewer UDAS's concern.

**Audience:**

Yes

**Audience Explanation:**

The reviewers unanimously agree that the paper has its audience.

**Claims And Evidence:**

Yes

**Claims Explanation:**

The reviewers unanimously agree that the claims in the paper are well supported. There is a minor concern from the reviewer UDAS on the lack of theoretical analysis. The AE considered this as a minor concern given the empirical evidence presented in the paper.